# The human dorsal anterior cingulate facilitates acceptance of unfair offers and regulates inequity aversion

Shotaro Numano[1,2]*, Chris Frith[3], Masahiko Haruno [1,2]*

**1** Center for Information and Neural Networks, Suita, Osaka, Japan, **2** Graduate School of Frontier Biosciences, Osaka University, Suita, Osaka, Japan, **3** Institute of Philosophy, School of Advanced Study, University of London, London, United Kingdom

\* shotaro.numano@gmail.com (SN); mharuno@nict.go.jp (MH)

## Abstract

Bargaining is a fundamental social behavior in which individuals often accept unfair offers. Traditional behavioral models, based solely on choice data, typically interpret such acceptance as simple reward-maximization. However, regulating inequity aversion may also play a critical role in these decisions. Incorporating response time alongside choice data allows us to quantify participants' internal mental effort when deciding to accept unfair offers. We conducted functional magnetic resonance imaging (fMRI) while participants played the ultimatum game, deciding within 10 s whether to accept or reject monetary offers from a proposer. Using the drift diffusion model (DDM), we quantified decision-making dynamics based on both choice and response time. Participants with lower levels of behavioral disadvantageous inequity (DI) aversion (reflected by a lower DDM weight for DI) showed stronger neural representations of DI in the dorsal anterior cingulate cortex (dACC). Functional connectivity analysis revealed a negative correlation between the dACC and bilateral ventrolateral prefrontal cortex (vlPFC) when DI was high. This connectivity predicted both rejection rates and response times associated with the acceptance of DI offers. Furthermore, robust linear regression showed that only dACC-vlPFC connectivity—not reward-related activity—explained both the rejection rates and response times. Finally, right vlPFC activity correlated with amygdala activity under high DI conditions. These findings suggest that the dACC plays a key role in regulating responses to DI, thereby facilitating the acceptance of unfair offers.

## Introduction

Bargaining is a crucial aspect of human interaction and identifying its cognitive and neural mechanisms is a major objective in human biology. In previous research, a key question for bargaining has been why people reject unfair offers even though accepting any offer is advantageous in terms of their payoff [1,2]. More specifically,

**Data availability statement:** All relevant data for replication are within the paper and its Supporting information files. Full data are available from the CiNet Institutional Data Access Committee (info@cinet.jp) for researchers who meet the criteria for access to confidential data. We used basic functions of Python in our data analysis.

**Funding:** This work was supported by KAKENNHI (22H05155, https://kaken.nii.ac.jp/ja/grant/KAKENHI-PLANNED-22H05155/), JST CREST (JPMJCR22P4, https://projectdb.jst.go.jp/grant/JST-PROJECT-22712882/), and JST Moonshot R&D (JPMJMS2011, https://projectdb.jst.go.jp/grant/JST-PROJECT-20339237/) and (JPMJMS259G, https://www.jst.go.jp/moonshot/en/program/goal9/) to M.H. The funders had no role in the study design, data collection and analysis, decision to publish or preparation of the manuscript.

**Competing interests:** The authors have declared that no competing interests exist.

**Abbreviations:** ACC, anterior cingulate cortex; AI, advantageous inequity; ANOVA, analysis of variance; BIC, Bayesian information criterion; BOLD, blood oxygenation level dependent; dACC, dorsal anterior cingulate cortex; DDM, drift-diffusion model; DI, disadvantageous inequity; EPI, echoplanar imaging; FWE, family-wise error; FWHM, full width at half maximum; fMRI, functional magnetic resonance imaging; FOV, field of view; GLM, general linear model; MCMC, Markov chain Monte Carlo; MNI, Montreal Neurologic Institute; OR, other-reward; PPI, psycho-physiological interaction; PSTH, peristimulus time histogram; ROI, region of interest; SR, self-reward; TE, echo time; TR, repetition time; VIF, variance inflation factor; vlPFC, ventrolateral prefrontal cortex; VOI, volume of interest; VS, voxel size; WAIC, widely applicable information criterion.

several studies reported that the insular cortex is responsive to inequity and contributes to the rejection of unfair offers [3–6]. Other studies have shown that subcortical structures, such as the amygdala and ventral striatum, also play a key role in representing inequity [7–9] and choosing rejection [8,10–12]. However, much less attention has been paid to the acceptance of unfair offers, which has been regarded mainly as simple reward-maximization [13].

Traditional behavioral studies based on choice (i.e., accepting or rejecting) alone tend to favor reward-maximization models [13] because these models can succinctly explain observed behavior [13,14]. However, unfair offers may evoke negative feelings associated with aversion to iniquity. For example, an unfair or disadvantageous offer can be taken as disrespectful and an offence to a responder. The rejection of unfair offers has been considered a form of costly punishment (altruistic punishment) for the unjust actions of others [15,16], and rejection behavior was reported to be driven by resentment or moral anger at the injustice [17]. It is therefore plausible that regulating emotions plays a key role in accepting unfair offers.

The consideration of response time alongside choice data may provide valuable information for revealing a participants' internal effort in deciding against inequity aversion and choosing to accept unfair offers. Several previous studies have categorized response times as fast or slow [18–21] and analyzed them in the context of dual-process theory [22]. Fast responses are often linked to intuitive decision-making, whereas slow responses are associated with more deliberate thought processes. However, there are limitations in such interpretations. It is difficult to determine whether slow responses stem from a type (i.e., slow or fast) of decision-making process or from conflicts between competing options [21]. Response times may also be influenced by other variables, such as the difficulty in distinguishing between options [18]. Therefore, it is likely more constructive to consider response time and choice simultaneously when examining participant's dynamic valuation process.

Sequential sampling models, particularly drift diffusion models (DDMs; Fig 2A) [23], are well-established computational models that estimate a participant's binary decision-making process by using both choice and response time. These models posit that individuals accumulate information incrementally during a trial and make a binary decision once enough evidence is gathered. The DDM has four parameters: decision boundary, relative starting point, nondecision time, and drift rate. The decision boundary represents the threshold of information required to make a choice. The relative starting point reflects the participant's prior response bias. Nondecision time refers to the duration spent on processes unrelated to decision-making, such as encoding the stimulus and executing a motor response like pressing a button. The drift rate reflects the participant's inclination toward one option (A or B) over time. Although originally introduced in behavioral psychology [24], DDMs have been utilized by neuroscientific studies for both animals and humans [25–28], and applied to a variety of tasks, ranging from perceptual decision-making [28–31] to social behavior [32,33].

In this study, we aimed to uncover the neural dynamics for accepting unfair offers. We conducted an functional magnetic resonance imaging (fMRI) experiment using



the ultimatum game. In each trial, participants decide to accept or reject a money-distribution offer from a proposer within 10 s. We analyzed behavior and fMRI data using a DDM. We assumed that the drift term relates to self-reward (SR) and to advantageous and disadvantageous inequity (advantageous inequity (AI) and disadvantageous inequity (DI), respectively). We hypothesized that if any regulation mechanism of DI-related negative feelings underlies the acceptance of unfair offers, we should observe some DI-correlated brain activity, which is large in people who exhibit a small DDM (drift) coefficient for DI. In other words, participants who do not reflect DI in their choice (i.e., they accept unfair offers) need long response times to decide against DI-related negative feelings in their choices. Based on previous studies on cognitive control (i.e., conflict detection and resolution) [34–40] and salient feelings with behavioral relevance, such as social and physical pain [41,42], we further hypothesized that the anterior cingulate cortex plays a crucial role in this process.

## Results

### Effects of DI on choices and response times

Healthy participants ($n = 63$) underwent an fMRI scan while playing an ultimatum game [13,14] as a responder (Fig 1A). In this game, proposers made a series of offers about money sharing to the responder, who decided whether to accept or reject each offer within 10 s. If the responder accepted the offer, the money was distributed as proposed; otherwise, neither side received any money. Each proposal consisted of a pair of rewards for the participant (SR) and the proposer (other-reward; OR).

Human behavior in the ultimatum game has been reported to be influenced by DI and AI (both defined based on SR and OR; see Computational models for inequity aversion in Materials and methods) [2,13,43]. We first confirmed that participant behaviors were consistent with prior findings regarding DI and AI and then examined the relationship between the choice and response time (Fig 1B).

The mean rejection rate for a disadvantageous offer (i.e., SR/OR < 1/1) was 44.0% (s.d. 33.1%), while that for an advantageous offer (i.e., SR/OR ≧ 1/1) was 2.81% (s.d. 9.48%). The difference between the two conditions was significant ($t(72) = 9.49, p = 2.58 \times 10^{-14}$, paired $t$ test). A repeated one-way analysis of variance (ANOVA) of the mean rejection rates yielded a significant main effect of the SR/OR rate ($F(6, 434) = 52.9, p < 7.60 \times 10^{-49}$). The mean rejection rates in the strongest (i.e., SR/OR = 1/9) and weakest (i.e., SR/OR = 2/3) DI trials were 73.5% (s.d. 39.1%) and 17.5% (s.d. 30.9%), with the former significantly higher than the latter ($t(117) = 8.93$ and $p = 7.00 \times 10^{-15}$, paired $t$ test). We also tested the contribution of SR, DI, and AI to the rejection rate using a generalized linear mixed model (GLMM) for all trials and found that only DI contributed significantly (S1 Table). These results clarified that participants rejected more as DI increased.

We next examined the participants' response time (Fig 1C). The mean response time for disadvantageous offers was 2.56 s (s.d. 1.30 s) and for advantageous offers 1.91 s (s.d. 1.05 s). An ANOVA of the mean response time yielded significant results for the main effect of the SR/OR rate and participant's choice, and for their interaction term (Table 1). The mean response time for acceptance was significantly longer than the one for rejection of the strongly disadvantageous offers (i.e., SR/OR = 1/9 and 1/4; $t(123) = 3.48$ and $^{***}p = 6.82 \times 10^{-4}$, Welch's $t$ test). By contrast, for advantageous offers (i.e., SR/OR = 3/2 and 7/3), the mean response time was significantly longer for the rejection than for the acceptance ($t(12) = 4.34$ and $^{***}p = 0.00105$, Welch's $t$ test). These results confirmed that only for strongly disadvantageous offers were acceptances slower than rejections.

### Evaluation of the drift diffusion model

To explore the dynamic neural processes for accepting unfair offers, we constructed DDMs (Fig 2A; see Materials and methods) and selected the best one to predict behavior (Fig 2B). More specifically, we calculated the widely applicable information criterion (WAIC; [44,45]) which is a generalization of the Akaike information criterion (AIC) in Bayesian modeling. The best model with the lowest WAIC included three components in the drift term: SR + DI + AI (WAIC = $7.45 \times 10^3$).

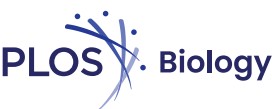

**Fig 1. Task and behavioral results. (A)** Task design. Participants performed the ultimatum game as a responder. In each trial, they chose to accept or reject the displayed proposal of money-distribution within 10 s. When the participant accepted the proposal, the money was distributed as proposed. Otherwise, neither received money. **(B)** Rejection rate for different proposals. Seven different ratios of self- and other-rewards (SR and OR) were used. The rejection rates increased in higher disadvantageous inequity (DI) conditions (e.g., 1/9), and it decreased in lower DI conditions (e.g., 1/1). The rejection rates also increased in higher advantageous inequity (AI) conditions (e.g., 2/3). **(C)** Response times in different conditions. With the increase in SR, the response time for acceptance decreased, while the one for rejection increased. In the highest DI condition (1/9), the mean response time for acceptance was slower than the one for rejection ($t(36) = 2.97$, Holm-Bonferroni corrected $^*p = 0.0370$, Welch's $t$ test). Numerical data are provided in S1 Data (A) and (C).

**Table 1. Repeated two-way ANOVA of the mean response time for each choice.**

| Variable | Sum of squares | df | *F* value | *P* value |
|---|---|---|---|---|
| Reward ratio | 94.2 | 6 | 12.5 | $3.59 \times 10^{-13}$ |
| Choice | 18.3 | 1 | 14.5 | $1.55 \times 10^{-4}$ |
| Reward ratio × Choice | 79.0 | 6 | 10.7 | $5.59 \times 10^{-11}$ |
| Residual | $6.68 \times 10^{2}$ | 531 | | |

We also found that the model that only included a constant in the drift term had poor predictive accuracy and the SR + DI model worked reasonably well, indicating that the DI term significantly improved the predictability. These results in Fig 2B also demonstrated that SR and OR are insufficient to account for choices and response times in the ultimatum game.

We confirmed generalization (prediction) performances of the DDM for different reward ratio trials (1/9, 1/4, 3/7, 2/3, 1/1, 3/2, and 7/3) by conducting cross validations of response times (Fig 2C; from left-top to right-bottom, Pearson's $r = 0.812$ and Holm-Bonferroni corrected $p = 3.91 \times 10^{-20}$, Pearson's $r = 0.7927$ and Holm-Bonferroni corrected $p = 8.48 \times 10^{-19}$, Pearson's $r = 0.7157$ and Holm-Bonferroni corrected $p = 2.891 \times 10^{-14}$, Pearson's $r = 0.8333$ and Holm-Bonferroni corrected $p = 6.01 \times 10^{-22}$, Pearson's $r = 0.8891$ and Holm-Bonferroni corrected $p = 1.63 \times 10^{-28}$, Pearson's $r = 0.9156$ and Holm-Bonferroni corrected $p = 5.24 \times 10^{-33}$, Pearson's $r = 0.9204$ and Holm-Bonferroni corrected $p = 6.09 \times 10^{-35}$, respectively). The findings show that the predicted response times closely matched the observed values, regardless of the reward ratio.

We conducted Markov chain Monte Carlo (MCMC) sampling in the model selection and parameter estimation processes (see Materials and methods) and obtained posterior distributions for the best model (Table 2). We checked whether these posterior distributions could reproduce participants' choices and response times. A repeated ANOVA of the posterior prediction of choice (predictive rejection rates; Fig 2D) confirmed a significant main effect of the reward ratio ($F(6, 434) = 54.5, p = 5.68 \times 10^{-50}$). We also examined the contribution of SR, DI, and AI to the predictive rejection rates using a GLMM and found a significant effect only for DI (S2 Table). Similarly, a repeated two-way ANOVA of the posterior prediction of response time (the predictive mean response time for each choice; Fig 2E) yielded significant effects for the main effects of the reward ratio and choice and for their interaction (Table 3). These results suggest that the best DDM model captured all key aspects of the participants' choices and response times.

Finally, we assessed the parameter recovery ability of our method by preparing two parameter sets of simulated agents, with one representing a reward-seeking agent and the other an inequity-aversive agent. We confirmed that the true parameter values were accurately recovered (Fig 2F and 2G, see also S2A Fig through S2D Fig for the parameter ranges that can be accurately estimated by our model; Numerical data are provided in S4 Data (C) through (F)).

We conducted a correlation analysis between individual drift parameters of the best model and the mean rejection rate and the mean response time of the same individual (Fig 3A–3C). $\beta$(DI), which represents the DDM coefficient for DI, showed the strongest correlation with the mean rejection rate (Fig 3A; Pearson's $r = 0.819$, Holm-Bonferroni corrected $p = 7.13 \times 10^{-16}$) and the mean response time for acceptance (Fig 3B; Pearson's $r = 0.274$, Holm-Bonferroni corrected $p = 0.0390$) and rejection (Fig 3C; $r = -0.449$ and Holm-Bonferroni corrected $p = 1.92 \times 10^{-3}$) in disadvantageous conditions.

We also tested whether this correlation is specific to the DI conditions by applying the same procedure to AI conditions. We found that $\beta$(DI) correlated with rejection rates (Pearson's $r = 0.333$, Holm-Bonferroni corrected $p = 0.0233$), while mean response times for acceptance and rejection did not show a correlation with $\beta$(DI). These results suggest that while the propensity to reject inequity generalizes across DI and AI trials for inequity-averse individuals, the processes underlying the decision speed are distinct.

We found that $\beta$(SR) correlated with the mean rejection rate (Fig 3D; Pearson's $r = -0.628$, Holm-Bonferroni corrected $p = 3.69 \times 10^{-8}$) and the mean response time for acceptance (Fig 3E; Pearson's $r = -0.344$ and Holm-Bonferroni corrected $p = 5.75 \times 10^{-3}$) but not with the mean response time for rejection (Fig 3F; Holm-Bonferroni corrected $p = 0.398$) in



**Fig 2. Model selection results and the relationship between drift parameters and behaviors. (A)** Drift diffusion model (DDM). The DDM contains four parameters: drift, nondecision time, boundary, and bias. Drift is assumed to be a linear combination of self-reward (SR), disadvantageous inequity (DI), and advantageous inequity (AI). We selected the best model using the WAIC. **(B)** Model selection results. The best drift term contained SR, DI, and

AI. Numerical data are provided in S2 Data. **(C)** Generalization performance of the best model. We evaluated generalization (prediction) performances by conducting cross validations of response times. The means of actual and predicted response times were compared for trials with different reward ratios: 1/9, 1/4, 3/7, 2/3, 1/1, 3/2, and 7/3 (Pearson's $r = 0.812$ and Holm-Bonferroni corrected $p = 3.91 \times 10^{-20}$, Pearson's $r = 0.7927$ and Holm-Bonferroni corrected $p = 8.48 \times 10^{-19}$, Pearson's $r = 0.7157$ and Holm-Bonferroni corrected $p = 2.891 \times 10^{-14}$, Pearson's $r = 0.8333$ and Holm-Bonferroni corrected $p = 6.01 \times 10^{-22}$, Pearson's $r = 0.8891$ and Holm-Bonferroni corrected $p = 1.63 \times 10^{-28}$, Pearson's $r = 0.9156$ and Holm-Bonferroni corrected $p = 5.24 \times 10^{-33}$, Pearson's $r = 0.9204$ and Holm-Bonferroni corrected $p = 6.09 \times 10^{-35}$, respectively). Numerical data are provided in S3 Data. We also simulated behaviors using the best model for **(D)** rejection rates and **(E)** mean response times for acceptance and rejection in different conditions. Numerical data are provided in S1 Data **(B)** and **(D)**. Parameter recovery tests with a simulated reward-seeker **(F)** and a simulated inequity avoider **(G)**. Each black line represents a true value, and each dot represents a recovered parameter from single inference process (this process was repeated 100 times for each parameter). Numerical data are provided in S4 Data (A) and (B).

**Table 2. Estimation of the DDM parameters. The mean and standard deviation (S.D.) of the estimated DDM parameters are shown. The upper boundary of our DDM represents rejection.**

| Parameter | Mean | S.D. |
|---|---|---|
| Boundary | 2.16 | 0.680 |
| Bias | 0.543 | 0.0962 |
| Nondecision time [s] | 0.591 | 0.266 |
| $\beta$(AI) | 0.254 | 0.868 |
| $\beta$(DI) | 0.371 | 0.481 |
| $\beta$(SR) | −0.930 | 0.504 |

**Table 3. Repeated two-way ANOVA of the mean predictive response time for each choice.**

| Variable | Sum of squares | df | *F* value | *P* value |
|---|---|---|---|---|
| Reward ratio | 84.9 | 6 | 11.5 | $5.11 \times 10^{-12}$ |
| Choice | 12.2 | 1 | 9.90 | $1.76 \times 10^{-3}$ |
| Reward ratio × Choice | 42.6 | 6 | 5.77 | $8.26 \times 10^{-6}$ |
| Residual | $5.70 \times 10^2$ | 464 | | |

unfair (advantageous and disadvantageous) conditions. We also found that $\beta$(AI) correlated with the mean rejection rate (Fig 3G; Pearson's $r = -0.313$, Holm-Bonferroni corrected $p = 0.00378$) but not with the mean response time in advantageous conditions. These results were replicated in the simulated behaviors (S1 Fig; Numerical data are provided in S5 Data (B)).

## Participants with a smaller $\beta$(DI) showed larger dACC activity that correlated with DI

Having established that $\beta$(DI) accurately represents behavioral choices and response times in disadvantageous conditions, we used this parameter in our fMRI analysis. First, we conducted a parametric fMRI analysis (GLM2 in Materials and methods). Because we hypothesized that for disadvantageous offers, accepting behavior is realized by the regulation of an aversion to DI, we sought brain activity that correlated with DI at the individual-level and beta values of the group correlated with–$\beta$(DI). The assumption behind this analysis was that the regulation process should be stronger when DI is large and even stronger in participants who accept more disadvantageous offers (i.e., correlated with–$\beta$(DI)).

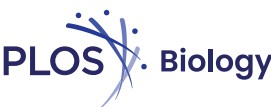

**Fig 3. Estimated drift parameters and behaviors** (A) **β(DI) and the rejection rate in DI conditions.** The rejection rate increased with increasing β(DI) (Pearson's $r = 0.819$, Holm-Bonferroni corrected $p = 7.13 \times 10^{-16}$). (B) β(DI) and mean response time for acceptance and (C) rejection in DI conditions. The mean response time for acceptance increased with increasing β(DI), while the mean response time for rejection decreased (Pearson's $r = 0.274$ and Holm-Bonferroni corrected $p = 0.0390$, $r = -0.449$ and Holm-Bonferroni corrected $p = 1.92 \times 10^{-3}$, respectively). (D) β(SR) and rejection rate in DI and AI (unfair) conditions. The rejection rate decreased as β(SR) increased (Pearson's $r = -0.628$, Holm-Bonferroni corrected $p = 3.69 \times 10^{-8}$). **(E)** β(SR) and mean response time for acceptance and (F) for rejection in unfair conditions. The mean response time for acceptance decreased with increasing β(SR) (Pearson's $r = -0.344$, Holm-Bonferroni corrected $p = 5.75 \times 10^{-3}$), but the mean response time for rejection had no correlation. (G) β(AI) and rejection rate in AI conditions. The rejection rate increased as β(AI) increased (Pearson's $r = -0.313$, Holm-Bonferroni corrected $p = 0.00378$). All numerical data are provided in S5 Data (A).

This analysis identified activity in the dACC (Fig 4A; $p < 0.05$ cluster-level FWE corrected; all activities are listed in (Table 4). Notably, we obtained this result by controlling for participant's response time (see Materials and methods): the neural contrast for response times did not produce significant activity ($p < 0.05$ cluster-level FWE corrected). We confirmed that the beta value for the dACC associated with DI was negatively correlated with β(DI) (Fig 4C; Pearson's $r = -0.549$, Holm-Bonferroni corrected $p = 3.21 \times 10^{-6}$). We also tested whether dACC activity was correlated with response time and/ or the evidence accumulation signal but found no significant correlation.

We further examined whether dACC activity survived after controlling for choices. We added a regressor which presents the participant's choice to individual-level GLM design (GLM2). Since 13 participants accepted all offers (with variable response times), we excluded those participants from this analysis. We obtained highly consistent dACC activity (S3 Fig and

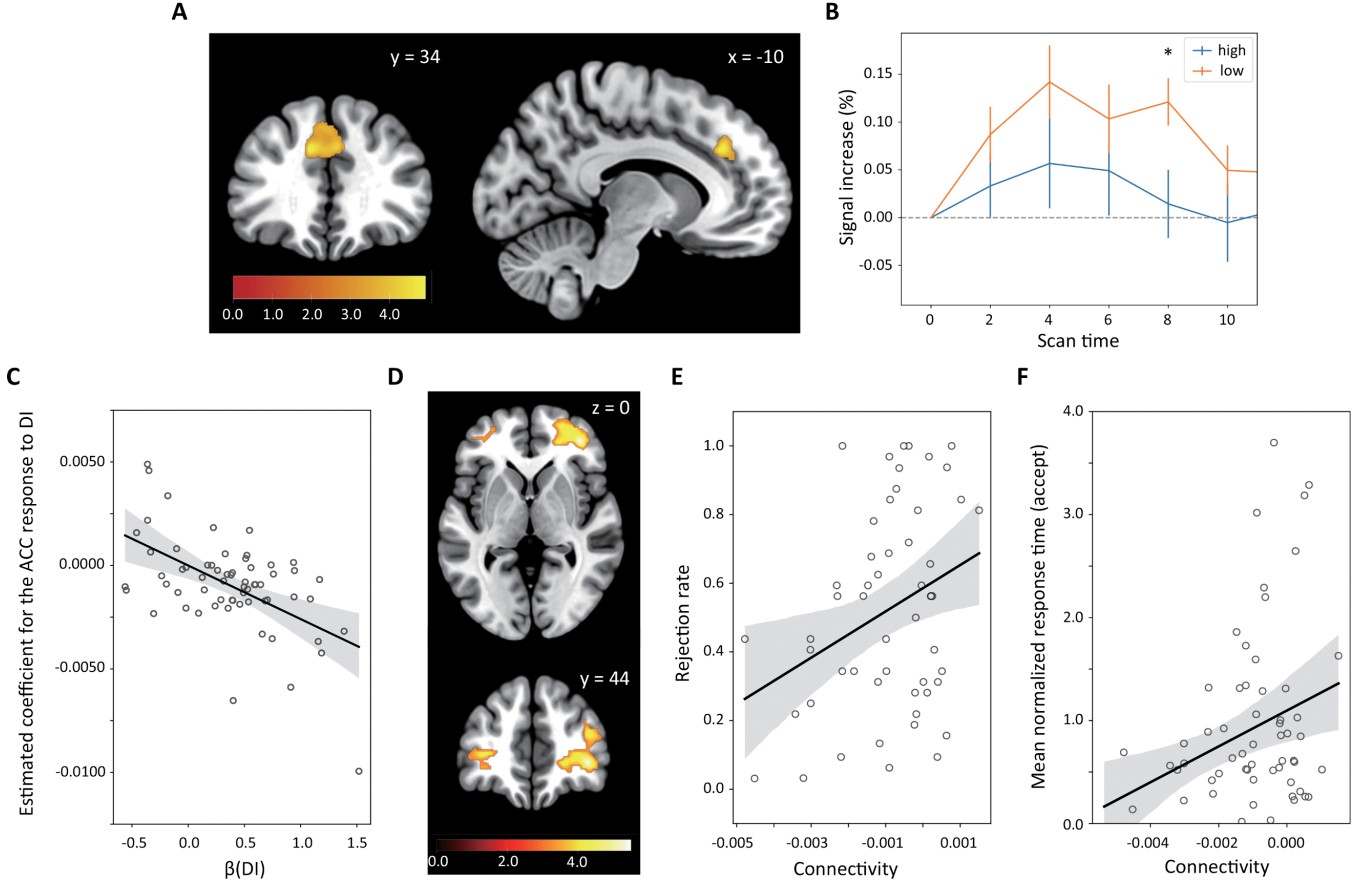

**Fig 4. Neural activity correlated with DI and psychophysiological interaction (PPI) analysis.** (A) Neural activity correlated with DI weighted by–$\beta$(DI). We found neural activity in the dorsal ACC (MNI coordinate = [−10, 34, 32]; cluster-level family-wised error (FWE) corrected $p < 0.05$). Numerical data are provided in S6 Data (A). (B) BOLD signal time series of the dACC. The BOLD signal time series at the peak (MNI coordinate = [−10, 34, 32]) showed a larger increase in the one half of participants with low $\beta$(DI)s than the other half across time. Specifically, we conducted a repeated two-way ANOVA of the signal increase at the dACC peak based on Time and Group (participant groups with high and low $\beta$(DI)s) (Table 5). We found a significant main effect of Group ($F(5,1,62) = 8.98$, $p = 0.00291$). We also found a significant difference in the PSTH at 8 s (Holm-Bonferroni corrected *$p = 0.0440$). Numerical data are provided in S6 Data (B). (C) The dACC responses and $\beta$(DI). We found that the beta value for the dACC associated with DI was negatively correlated with $\beta$(DI) (Pearson's $r = -0.549$, Holm-Bonferroni corrected $p = 3.21 \times 10^{-6}$). Numerical data are provided in S6 Data (D). (D) Functional connectivity between the ACC and vlPFC. The vlPFC (MNI coordinate = [38, 44, 0]; cluster-level FWE corrected $p < 0.05$) showed a negative interaction with the ACC when DI was large (see also Table 6). Numerical data are provided in S6 Data (C). (E) Relationship between connectivity and rejection rates. The mean rejection rate decreased as connectivity became negative (Pearson's $r = 0.318$, Holm-Bonferroni corrected $p = 0.0402$). (F) Connectivity and mean normalized response time for acceptance in DI conditions. The normalized response time for acceptance decreased (became faster) as connectivity became negative (Pearson's $r = 0.269$, Holm-Bonferroni corrected $p = 0.0391$). Numerical data of (E) and (F) are provided in S6 Data (D).

S3 Table; cluster-level family-wised error (FWE) corrected $p < 0.05$ and numerical data are provided in S7 Data). These results suggest that the dACC responds to DI but not to response time, general conflict, evidence accumulation signal, or choice [46].

Since activity in the dACC is correlated with–$\beta$(DI), the time series of neural activity in this region should differ according to the $\beta$(DI). To confirm this hypothesis, we compared the BOLD signal time series of this region between the one half of participants with high $\beta$(DI)s and the other half with low $\beta$(DI)s by conducting a repeated two-way ANOVA of the signal increase at the dACC (Table 5). We found a significant main effect of Group ($F(5,1,62) = 8.98$, $p = 0.00291$). We also found a significant difference in the peri-stimulus time histogram (PSTH) at 8 s (Holm-Bonferroni corrected *$p = 0.0440$). These results demonstrated that participants with low $\beta$(DI)s exhibited larger dACC activity than participants with high $\beta$(DI)s (Fig 4B).

**Table 4. Neural activities to DI weighted by–β (DI). We performed cluster-level family-wised error correction (p < 0.05). All clusters are displayed.**

| MNI coordinate (mm) | | | ROI name | | T-value | Cluster-level p-value (FWE corrected) | Cluster size (mm³) |
|---|---|---|---|---|---|---|---|
| X | Y | Z | | | | | |
| −48 | 16 | 4 | Frontal Inf Tri | L | 5.79 | 0.00560 | 2,104 |
| −54 | −38 | 52 | Parietal Inf | L | 5.34 | $5.02 \times 10^{-6}$ | 5,576 |
| 54 | −44 | 52 | Parietal Inf | R | 5.27 | $3.99 \times 10^{-5}$ | 4,440 |
| −42 | 48 | 12 | Ventrolateral Prefrontal Cortex | L | 5.09 | 0.0118 | 1,808 |
| −10 | 34 | 32 | Anterior Cingulate Cortex | L | 4.52 | $1.58 \times 10^{-5}$ | 4,936 |
| 40 | 22 | 52 | Dorsolateral Prefrontal Cortex | R | 4.77 | 0.00510 | 2,144 |
| 2 | −38 | 30 | Posterior Cingulate Cortex | L | 4.24 | 0.00980 | 1,880 |
| −22 | 24 | 46 | Frontal Mid 2 | L | 3.77 | 0.0145 | 1,728 |

**Table 5. Repeated two-way ANOVA of the signal increase at the dACC peak based on the Time and Group (participant groups with large and low β(DI)s).** We found a significant main effect of Group (p = 0.0029).

| Variable | Sum of squares | df | F value | P value |
|---|---|---|---|---|
| Time | 0.445 | 5 | 2.46 | 0.0328 |
| Group | 0.325 | 1 | 8.98 | 0.00291 |
| Time × Group | 0.105 | 5 | 0.583 | 0.713 |
| Residual | 13.2 | 366 | | |

## dACC-vlPFC interaction encodes the rejection rate and response time for accepting DI offers

To further examine the regulation mechanism by the dACC (Fig 4A and Table 4), we conducted a psychophysiological interaction (PPI) analysis using the gPPI toolkit [47]. We defined a volume of interest from the peak positions of each cingulate cluster and ran the PPI analysis by setting these regions as the seed. We looked for significant negative interactions when DI was large.

We found a significant negative interaction between the dACC (Fig 4A and Table 4; MNI coordinates = [−10, 34, 32]) and the vlPFC (Figs 4D, S4, and Table 6; cluster-level FWE corrected p < 0.05; MNI coordinates = [38, 44, 0]) when DI was large. Notably, we did not use any DDM-derived parameters to reveal this connectivity because such use may produce false-positive correlations with response times. Our results suggest that the dACC regulates the vlPFC when DI is large. We did not find any significant connectivity to other seeds (Table 4).

**Table 6. Neural activity identified by the PPI analysis whose seed ROI is the dACC. We performed cluster-level family-wised error correction (p < 0.05). All clusters are displayed.**

| MNI coordinate [mm] | | | ROI name | | T-value | Cluster-level p-value (FWE corrected) | Cluster size [mm^3] |
|---|---|---|---|---|---|---|---|
| X | Y | Z | | | | | |
| 38 | 44 | 0 | Ventrolateral prefrontal cortex | R | 5.58 | $2.22 \times 10^{-7}$ | 5,352 |
| −58 | −22 | 38 | Supramarginal gyrus | L | 5.07 | 0.00174 | 1928 |
| −34 | 48 | 6 | Ventrolateral prefrontal cortex | L | 4.81 | 0.0160 | 1,272 |
| −36 | −32 | 38 | Inferior parietal lobule | L | 4.52 | 0.00307 | 1,752 |

We next examined if this interaction between the dACC and the vlPFC encodes the rejection rate and the response times for accepting disadvantageous offers. We found that dACC-vlPFC connectivity exhibits a significant correlation with the rejection rate (Fig 4E; Pearson's $r = 0.318$ and Holm-Bonferroni corrected $p = 0.0402$) and the normalized response time for accepting disadvantageous offers (Fig 4F; Pearson's $r = 0.269$ and Holm-Bonferroni corrected $p = 0.0391$). In addition, we observed that dACC-vlPFC connectivity has a significant correlation with the raw response time for acceptance (Pearson's $r = -0.340$ and Holm-Bonferroni corrected $p = 0.00283$). We did not find any significant correlation with rejection rates and response times for other structures (Table 6). These results suggest that regulation of the vlPFC via the dACC underlies the decision to accept DI offers.

**Activity in vlPFC and amygdala synchronizes when DI is large**

The vlPFC has anatomical and functional connections to subcortical regions such as the amygdala and is known to be involved in emotion regulation [48–50]. The amygdala was reported to respond to inequity in people who dislike it [7], and we confirmed significant left amygdala activity correlated with DI (GLM1 design) in the half of participants who had a high $\beta$(DI) (S5 Fig and S4 Table; cluster-level FWE corrected $p < 0.05$ with small volume correction and numerical data are provided in S8 Data (B)). Based on these observations, we hypothesized that the vlPFC has strong functional connectivity to amygdala at the offer timing when DI is large and performed a PPI analysis.

We defined the peak positions of the vlPFC (MNI coordinates = [38, 44, 0]) as a volume of interest, and found an interaction between the vlPFC (Fig 4D) and the amygdala (Fig 5A and Table 7; cluster-level FWE corrected $p < 0.05$ with small

**A**

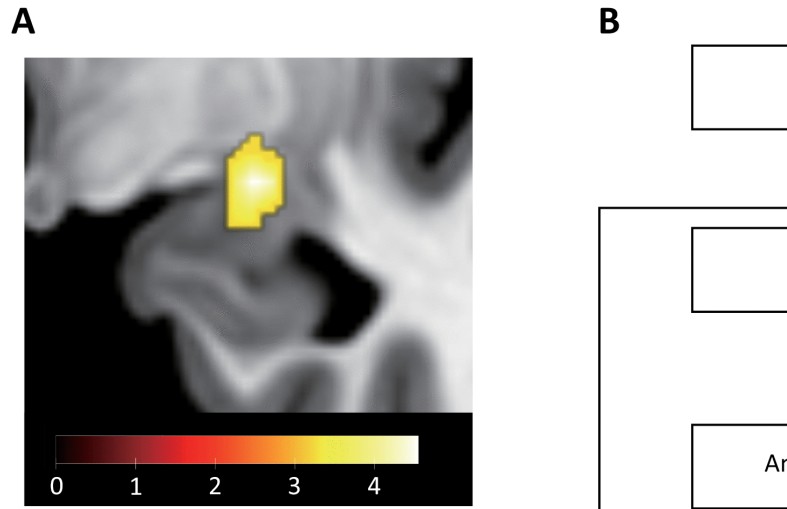

**B**

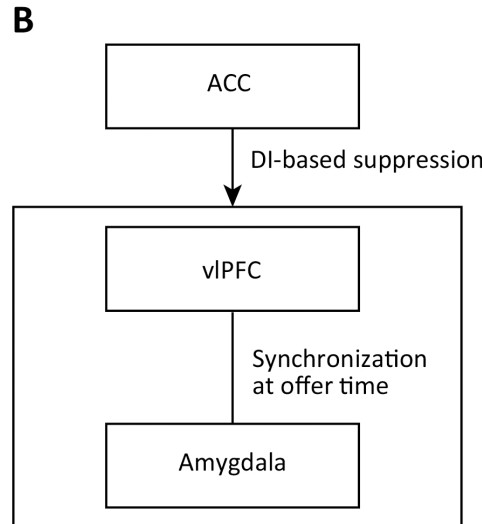

**Fig 5. Psychophysiological interaction analysis of the vlPFC. (A)** Functional connectivity of the vlPFC cluster. The amygdala (MNI coordinate = [24, −8, 12]; cluster-level FWE corrected $p < 0.05$ with small volume correction) had a positive interaction with the vlPFC cluster when DI was large. The figure only displays the peak cluster in the amygdala (see also Table 7). Numerical data are provided in S8 Data (A). **(B)** Conceptual diagram of our findings.

**Table 7. Neural activity identified by the PPI analysis whose seed ROI is the vlPFC. We conducted small volume correction for multiple comparisons (cluster-level FWE corrected $p < 0.05$) with the hypothesis that the bilateral amygdala is involved.**

| MNI coordinate (mm) | | | ROI name | | T-value | Cluster-level p-value (FWE corrected) | Cluster size (mm³) |
|---|---|---|---|---|---|---|---|
| X | Y | Z | | | | | |
| 24 | −8 | −12 | Amygdala | R | 4.73 | 0.0189 | 64 |

volume correction; MNI coordinates = [24, −8, −12]). In contrast to connectivity between the dACC and the vlPFC (Fig 4E and 4F), this interaction did not show a correlation with the mean rejection rate and the mean normalized response time for acceptance. Therefore, we concluded that activity in the vlPFC and the amygdala was synchronous at the offer timing when DI was large (Fig 5B).

A similar regulation mechanism to DI may also work for SR. We therefore conducted an analysis where individual activity correlated with SR was weighted with–$\beta$(SR) in the group-level (GLM5). We did not find significant activity (we only found neural activity in the dACC with a very weak threshold without multiple-comparison correction, uncorrected $p < 0.01$: S6 Fig and S5 Table; Numerical data are provided in S9 Data). Furthermore, this region did not show any significant connectivity with any other region (uncorrected $p < 0.001$: only two clusters survived with a very low threshold: uncorrected $p < 0.01$ with a cluster-forming voxel threshold > 40 mm³, S6 Table). Similarly, we tested AI by applying the individual-level results correlated with AI weighted with–$\beta$(AI) in the second-level analysis and found no significant correlation ($p < 0.05$ cluster-level FWE corrected).

We have shown so far that negative functional connectivity between the dACC and vlPFC plays a crucial role in accepting unfair offers. An intriguing question then is the relative contribution between this connectivity and reward-correlated brain activity. To answer this, we conducted a robust linear regression analysis of the rejection rate and response time based on beta values for SR-correlated activities and the DI-correlated interaction between the dACC and vlPFC. More specifically, we extracted beta values from the peaks of clusters (uncorrected $p < 0.005$, individual SR-correlated activity weighted by $\beta$(SR) in the group-level analysis, S7 Table) and PPI results. As a result, only connectivity between the dACC and vlPFC showed a significant effect, explaining both the mean rejection rate (S8 Table) and the normalized response time (S9 Table). These results suggest that regulating responses to DI by connectivity between the dACC and vlPFC plays a more crucial role in accepting unfair offers than previously thought.

## Comparing DDM and standard value-based models

Finally, we compared the DDM and a standard value-based model (i.e., a logistic regression model) that considers choice but not response time. We analyzed participants' behavior and neural activity using a logistic regression model [51]. For this purpose, we assumed that the best predictive logistic regression model has the same three parametric components as the best DDM model.

We estimated the coefficient $\gamma$ of the logistic regression model using $R$ [52] and found a significant correlation between $\gamma$(DI) and the mean rejection rate (Pearson's $r = 0.740$ and $p = 0.00328$). However, the correlation between $\gamma$(DI) and mean response time was not significant (Pearson's $r = -0.0971$ and $p = 0.445$). Furthermore, $\gamma$(DI) did not show a positive correlation with the difference in mean response times for acceptance and rejection (Pearson's $r = 0.0442$ and $p = 0.731$, Pearson's $r = -0.312$, and $p = 0.345$, respectively), which is in contrast to $\beta$(DI) (Fig 3B and 3C). These data confirmed that only the DDM can capture behaviors in the time domain.

Observing that only $\beta$(DI), but not $\gamma$(DI), encodes response times, we compared the two parameters in the fMRI analysis. More concretely, we performed a GLM analysis, where all second-level regressors were replaced by the coefficients of the logistic model (GLM3). In sharp contrast with Fig 4A, we identified much weaker activity in the dACC (Fig 6 and Table 8; $p < 0.001$ uncorrected, neural activity in the dACC area is shown in bold in Table 8). We conducted another GLM analysis, where the second-level regressors contained both $\beta$(DI) and $\gamma$(DI) (GLM4) and contrasted $\beta$(DI) minus $\gamma$(DI) to evaluate the relative power of these variables. This analysis identified clusters in the dACC with a moderate threshold (S7 Fig; uncorrected $p < 0.05$; Numerical data are provided in S10 Data (B)). These results suggest that the use of the DDM, i.e., considering both choice and response time, is crucial for revealing dACC activity in the acceptance of unfair offers.

## Discussion

The main purpose of this study was to test whether regulating inequity aversion plays a crucial role in accepting unfair offers and, if it does, to identify the underlying neural substrates. We conducted an fMRI experiment of the ultimatum



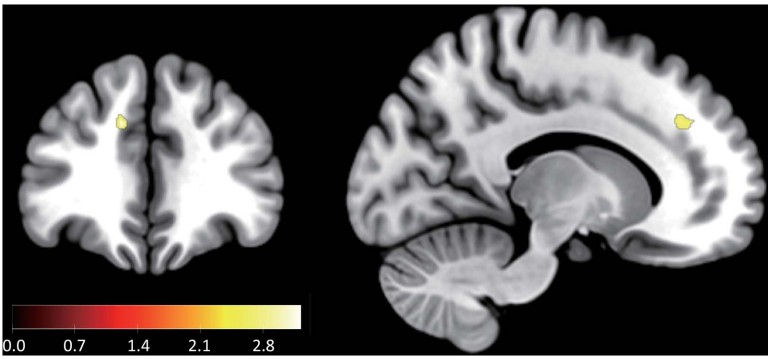

**Fig 6. Neural activity correlated with the coefficient of the standard value-based model.** Only neural activity in the ACC (MNI coordinate = [−12, 36, 32]; uncorrected $p < 0.001$; see also Table 8) and a peak cluster in the dACC were found ($p < 0.005$ uncorrected for display purposes). Numerical data are provided in S10 Data (A).

**Table 8. Neural activity to DI weighted by $\gamma$–(DI) (the value-based model). We set the statistical threshold for significance at $p < 0.001$ uncorrected for multiple comparisons. All clusters are displayed. Neural activity in the dACC area is shown in bold (Peak MNI coordinate = [−12, 36, 32]).**

| MNI coordinate [mm] | | | ROI name | | T-value | Peak-level p-value (uncorrected) | Cluster size [mm³] |
|---|---|---|---|---|---|---|---|
| X | Y | Z | | | | | |
| −26 | −98 | 0 | Middle occipital gyrus | L | 4.59 | $1.15 \times 10^{-5}$ | 872 |
| 24 | −96 | −6 | Inferior occipital gyrus | R | 4.51 | $1.53 \times 10^{-5}$ | 952 |
| −38 | 54 | 18 | Middle frontal gyrus | L | 3.98 | $9.49 \times 10^{-5}$ | 72 |
| 18 | 28 | 54 | Medial prefrontal cortex | R | 3.76 | 0.000193 | 160 |
| 52 | −54 | 50 | Inferior parietal lobule | R | 3.62 | 0.000303 | 96 |
| −50 | −48 | 40 | Inferior parietal lobule | L | 3.54 | 0.000389 | 120 |
| 24 | −30 | −4 | Hippocampus | R | 3.37 | 0.000658 | 16 |
| **−12** | **36** | **32** | **Anterior cingulate cortex** | **L** | **3.34** | **0.000719** | **8** |
| 40 | 40 | 36 | Dorsolateral prefrontal cortex | R | 3.24 | 0.000972 | 8 |

game and analyzed behavioral and fMRI data using a DDM that infers internal representations to model both behavioral choices and response times. We found that participants who exhibited a smaller DDM parameter for DI (i.e., $\beta$(DI)) and dampened disadvantage-driven rejection (i.e., higher acceptance rate) showed heightened dACC activity for DI. Our PPI analysis revealed that, when unfair offers are presented, the dACC exhibits negative functional connectivity with the vlPFC. Importantly, we demonstrated that this connectivity between the dACC and vlPFC encodes the average rejection rate and the response time for accepting unfair offers. Robust linear regression also showed that dACC-vlPFC connectivity—not reward-related activity—explains both rejection rates and response times. We also found that the vlPFC has synchronized activity with the amygdala at offer time when unfair offers are presented. In contrast, a conventional value-based system (i.e., logistic regression) that only considers choices did not detect these fMRI signals, suggesting that consideration of both the choice and response time is necessary to elucidate the neural mechanism for accepting unfair offers.

Accepting unfair offers has been regarded as a reward-maximization process [13,53–55] since behavioral studies based on choice alone favor the simplest model to succinctly explain observed behaviors and suit well the definition of "economic man" [13,14,54,55]. However, the results of the present study suggested that a more complex regulation mechanism is working in the human brain to choose a reward-maximizing option. Notably, not only the integration of response



times and choices in the framework of the DDM but also the fMRI analysis based on DDM parameters were central to the present findings. However, it is worth noting that a comparison between the DDM and logistic regression only in terms of choices showed that the explanatory ability of the latter was better (WAIC values: $7.45 \times 10^3$ for DDM and $1.75 \times 10^3$ for logistic regression), revealing complementary advantages of the two approaches.

We found that the connectivity between the dACC and the vlPFC, but not between the vlPFC and the amygdala (Fig 5A), encodes both the mean rejection rate and response time for acceptance (Fig 4F), showing that the dACC and the vlPFC play a central role in deciding to accept. These results are consistent with previous reports that found that dACC plays a pivotal role in cognitive control [34,39,40,56,57] and emotion regulation [38,41,42,49,58,59] and also that the dACC has anatomical projections to the vlPFC [48,49,60].

We found that the vlPFC exhibited synchronized activity with the amygdala at the time of unfair offers. This observation is consistent with previous findings showing anatomical and functional connections between the vlPFC and amygdala [38,48,49,59,61]. Given that the amygdala has been reported to encode economic inequity [7–9] and contribute to the rejection of unfair offers [8,10–12], our results suggest that the dACC and vlPFC may regulate negative affective responses to inequity in the amygdala through their coordinated interactions.

Notably, amygdala activity correlated parametrically with DI only in participants who exhibited the strongest behavioral DI aversion (i.e., those with the highest $\beta(DI)$), but not in those with weaker aversion. This pattern is consistent with a previous fMRI study showing that prosocial individuals—who strongly dislike inequity and prioritize joint outcomes—displayed a relationship between amygdala activity and inequity, whereas more individualistic individuals did not [7]. Together, these findings suggest a potentially nonlinear relationship between inequity aversion and amygdala responses, in which regulatory mechanisms involving the dACC and vlPFC may also contribute. However, further investigation is required to confirm this interpretation, as we did not directly assess social value orientation [7] in the present study.

This study mainly focused on the regulation of DI-related negative feelings to achieve the acceptance of unfair (DI) offers. It is also possible that a similar regulation mechanism for SR is involved in rejecting all kinds of unfair offers (DI and AI). To address this, we conducted an fMRI analysis where we looked for SR-correlated activity weighted by–$\beta(SR)$. The intuition here is that when SR is large, some regulation mechanism works to achieve rejection in people who did not take account SR into their selection of rejection. However, we observed neither significant activity nor correlation between rejection behavior and (nonsignificant) weak brain activity (S6 Fig). This result may highlight a unique role of DI in human social decision-making.

Related to this, our robust linear regression analysis revealed that ACC-vlPFC connectivity but not reward (SR)-related activity predict the mean rejection rate (S8 Table) and the mean normalized response time for acceptance (S9 Table). Although we need to carefully consider the possibility that this result becomes apparent particularly in the ultimatum game or with fMRI signals, this suggests that the regulation of DI plays a more crucial role in accepting unfair offers than previously thought.

There are several limitations in this study. We first note that our echo-planar imaging (EPI) sequences resulted in partial signal dropout in the orbitofrontal cortex (OFC; S9 Fig), and our study cannot speak directly to OFC contributions. Second, the number of trials in the AI condition was too small to detect brain activity robustly (only 14 trials in the whole task). Because AI is also important for human social decision-making [62–65], further investigation of AI is necessary in comparison with the mechanisms identified in the present study. Third, although we successfully extracted essential contributions of the dACC and vlPFC, we need to prove direct causality in these structures. This may be addressed by noninvasive stimulation techniques such as transcranial magnetic stimulation [66,67] and transcranial focused ultrasound stimulation [68]. Lastly, although the DDM considering both choice and response time was central to our findings [23], future studies should design more sophisticated computational models that explain regulation processes during decision-making directly and mechanistically. Further development of such computational models would help establish more comprehensive understanding of human decision-making processes.

Finally, the results of the present study may help to link bargaining behavior and social problems such as mood disorders. For instance, previous research reveals that people with depression show a greater emotional response to unfair

offers [69] and have a greater aversion to inequity [12,70]. Furthermore, people with major depression show a lower acceptance rate for unfair offers [71], while in healthy volunteers, unfair offers induce sadness to cause a lower acceptance rate for such offers [72]. Our results suggest that such symptoms may be linked to the regulation of emotional responses to unfair offers.

## Materials and methods

### Participants

We initially enlisted 71 participants for the experiment but excluded 8 due to significant or erratic head movements (greater than 1 mm) in their fMRI scans. As a result, 63 participants (41 males, 22 females; average age = 22.0, standard deviation = 3.27) were included in the main experiment. Thirteen participants accepted all offers (but with variable response times). So, these subjects were excluded from analyses that involved within subject comparisons of accepted and unaccepted offers. This study adhered to the Declaration of Helsinki and was approved by the ethics committee of the National Institute of Information and Communications Technology (approval number: B210142205). All participants provided written informed consent.

### Experiment procedure

Participants began by reading a document outlining the task instructions and completed a consent form. They were informed that all proposals would be made by students from local universities, and that both the proposer and the proposal would vary in each trial. They were also told that their responses would affect the monetary outcomes for both the proposer and the responder. Additionally, participants were instructed to make their decisions within 10 s and press the corresponding button. Before starting the main task, they familiarized themselves with the time limit (<10 s) and task setup through three practice trials.

### Task

Participants played the ultimatum game (Fig 1A) as responders. They assessed proposals from an anonymous proposer and decided whether to accept or reject the offer by pressing a button within 10 s. If the participant did not press the button within then, the error trial was excluded from the analysis. If the responder accepted the offer, the money was allocated as proposed. If the responder rejected the offer, neither the proposer nor the responder received any money.

The base offers were one of the following: ¥350–150, ¥300–200, ¥250–250, ¥200–300, ¥150–350, ¥100–400, or ¥50–450 for the responder (participant) and the proposer. In each trial, a random number between ¥−25 and ¥25 was added to each reward, creating variation in the offers displayed. This design was intended to encourage participants to think critically and prevent pattern-based responses. Each base offer had eight trials, totaling 56 trials for the entire task (S10 Table).

Each trial began with 3 s of fixation, followed by a 1-s display of the proposer's face and name. Participants then had 10 s to decide whether to accept or reject the proposal (choice period in Fig 1A). After making a decision, participants saw their choice displayed for 1 s ("Feedback" phase in Fig 1A). After the feedback, there was a 6.5-s rest period before the next trial. Each trial lasted between 22.5 and 24.5 s, with the entire task taking 22 min and 16 s. During the main experiment, the viewing distance was between 91.0 and 94.0 cm, the viewing angle was 21.7-22.4 degrees, and the screen luminance was 110.3 cd/m².

### Computational models for inequity aversion

We defined DI and AI for trial $t$ as follows:

$$DI_t = \max(\text{other-reward}_t - \text{self-reward}_t, 0)$$
$$AI_t = \max(\text{self-reward}_t - \text{other-reward}_t, 0)$$



where self-reward$_t$ (SR$_t$) represents the responder's (participant's) reward and other-reward$_t$ (OR$_t$) represents the proposer's reward. We confirmed that each parameter, DI, AI, and SR, has a variance inflation factor (VIF) of 1.29, 2.77, and 3.06, respectively which indicated low collinearity among these parameters (VIF > 10 is a standard criteria for collinearity; S11 Table) [73]. To better understand participants' behavior, we examined how DI, AI, and SR influenced the rejection of unfair offers and response times by applying two computational models: a DDM and a logistic regression model.

The DDM consists of four parameters: decision boundary, relative starting point, nondecision time, and drift rate ($\delta$) (Fig 2A; for more details, see also [74]). In this study, we linked rejection to the upper boundary of the DDM and acceptance to the lower boundary. We defined $\delta$ as:

$$\delta_t = \sum_{X \in 2^\Omega} \beta(X) \times X_t,$$

where $\beta(X)$ represents the weight of the variable X, and $2^\Omega$ refers to the power set of variables in the task, where $\Omega$ = {SR, AI, DI}. We estimated the DDM parameters using the HSSM package (https://lnccbrown.github.io/HSSM). This toolbox applies approximate hierarchical Bayesian inference to DDMs using Python and is an enhanced version of HDDM [43,44], which is commonly used for similar applications [23,75–78].

We specified a prior distribution for the DDM parameters as follows:

$$\alpha \sim \text{HalfNormal}(\sigma = 2.0),$$

$$b \sim \text{Uniform}\left(\text{lower} = 0.0, \ \text{upper} = 1.0\right),$$

$$\tau \sim \text{HalfNormal}(\sigma = 2.0),$$

$$\beta \sim \text{Uniform}\left(\text{lower} = -3.0, \ \text{upper} = 3.0\right),$$

where α represents boundary separation, b represents bias, τ represents nondecision time, and $\beta$ represents the linear coefficient in the drift term. MCMC sampling was performed with 4 chains and 15,000 burn-in iterations, and 5,000 posterior samples were obtained for each chain. We confirmed that Gelman-Rubin diagnostics for all samples were less than 1.1, which is generally used as a threshold to determine the convergence of Monte Carlo sampling [45].

By contrast, to implement the value-base model, we defined the subjective values for rejection, $V_{\{t, \text{reject}\}}$ and for acceptance, $V_{\{t, \text{accept}\}}$ at trial $t$. These values were integrated into a difference function, $v(t)$, for the participant as follows:

$$v(t) = V_{\{t, \text{reject}\}} - V_{\{t, \text{accept}\}}.$$

This value function was used to calculate the probability of choosing rejection, $p_{\text{reject}}(t)$, using the logistic function $p_{\text{reject}}(t) = 1 / \left[1 + \exp\left(-v(t)\right)\right]$.

To compare the DDM with the value-based logistic regression model, we assumed that the best predictive logistic model included the same components as the drift term in the best-performing DDM. Thus, we redefined the subjective values for rejection and acceptance as follows:

$$V_{\{t, \text{reject}\}} = \gamma_0 \times g_{\{t, \text{reject}\}}$$
$$V_{\{t, \text{accept}\}} = \gamma_0 \times g_{\{t, \text{accept}\}} - \gamma(\text{AI}) \times \text{AI} - \gamma(\text{DI}) \times \text{DI}$$

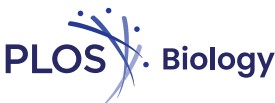

where, $g_{\{t,X\}}$ represents the participant's gain for option X, and $\gamma$ denotes a coefficient. In our task, $g_{\{t,\text{reject}\}}$ was 0, and $g_{\{t,\text{accept}\}}$ was $SR_t$. Logistic regression analysis was performed using the bias-reduction method provided by the Brglm package [79].

**Statistical analysis of behavior**

In the ultimatum game, inequity is known to affect choices [2,13,43]. We set choices in each trial as the objective variable (acceptance = 0 and rejection = 1) and investigated the contribution of payoff (SR), AI, and DI by logistic regression using a GLMM. We found significant contributions of every parameter to choices (S1 Table). We also confirmed this result in predictive choices (S2 Table).

**Model selection and evaluation**

Since we defined the drift rate $\delta_t$ as a linear combination of the task variables, we considered nine possible models:

$$\delta_t = \begin{cases} \beta\,(\text{SR}) \times \text{SR}_t \\ \beta\,(\text{AI}) \times \text{AI}_t \\ \beta\,(\text{DI}) \times \text{DI}_t \\ \beta\,(\text{SR}) \times \text{SR}_t + \beta\,(\text{AI}) \times \text{AI}_t \\ \beta\,(\text{SR}) \times \text{SR}_t + \beta\,(\text{DI}) \times \text{DI}_t \\ \beta\,(\text{AI}) \times \text{AI}_t + \beta\,(\text{DI}) \times \text{DI}_t \\ \beta\,(\text{SR}) \times \text{SR}_t + \beta\,(\text{OR}) \times \text{OR}_t \\ \beta\,(\text{SR}) \times \text{SR}_t + \beta\,(\text{AI}) \times \text{AI}_t + \beta\,(\text{DI}) \times \text{DI}_t \\ \text{Constant only.} \end{cases}$$

We compared the models using the WAIC [44,80], a standard criterion for MCMC sampling. We computed the WAIC for each trial and summed the results to identify the model with the lowest total, which was considered the best model. We then investigated the relationship between the drift parameters and behavior, i.e., the mean rejection rate and the mean response time for each choice. Although we focused on drift parameters in the main text (Fig 3), we also examined the relationship between behaviors and other DDM parameters such as boundary and bias (S8 Fig). We obtained similar results to the observations reported in our previous study [74] (see also [74] for more details).

**Model validation of the best model**

To assess the predictability (generalization ability) of the optimal model, we carried out a leave-one-out cross-validation procedure across the seven (reward) distribution conditions. In each fold, one trial was held-out per condition, model parameters were estimated by the remaining data, and the response time for the held-out trial was predicted. This process was repeated eight times, matching the number of trials in each distribution condition.

We also verified our estimation method via a parameter recovery test [81]. We prepared two ground-truth parameter sets (simulated participants): reward-seeking and inequality-averse (consisting of boundary separation, nondecision time, starting bias, $\beta$(DI), $\beta$(AI), and $\beta$(SR)). From each set, we simulated choices and reaction times in 56 trials and re-estimated the model parameters on these synthetic behavioral data. This simulation and recovery procedure was repeated 100 times.

**fMRI acquisition**

We assessed participants' neural activity during the task performance using fMRI. We acquired brain images with a 3T MRI (Prisma; Siemens Medical Systems) equipped with a 64-channel head coil at the Center for Information and Neural Networks, National Institute of Information and Communications Technology (Japan). To correct for distortion, we first obtained field maps covering the entire brain with the following parameters: repetition time (TR) = 753 ms, echo time 1

(TE1) = 5.16 ms, echo time 2 (TE2) = 7.62 ms, flip angle (FA) = 9°, field of view (FOV) = 256 mm, and voxel size (VS) = 2 × 2 × 2 mm. Next, we acquired T2*-weighted images during the task using a multi-echo-planar pulse sequence with TR = 2,000 ms, TE = 30 ms, FA = 75°, FOV = 200 mm, versus = 2 × 2 × 2 mm, and multi-band factor = 3. Finally, we obtained high-resolution T1-weighted images after the task session using the MPRAGE pulse sequence with TR = 1,900 ms, TE = 3.37 ms, FA = 9°, FOV = 256 mm, and versus = 1 × 1 × 1 mm.

To examine coverage in the EPI images, we compared the mean EPI images and T1-weighted images, which were co-registered and normalized (S9 Fig). We found that subcortical areas such as the amygdala were successfully imaged (S9B and S9C Fig), while there was partial dropout of OFC in the EPI images (S9A Fig).

### Image preprocessing

The images were processed using SPM12 (http://www.fil.ion.ucl.ac.uk/spm) within MATLAB 2018a. The preprocessing steps included slice-timing correction (with a reference time of 1,000 ms), motion correction, removal of movement-induced variance (unwarping) using a measured field map, co-registration to the T1 image, spatial normalization, and spatial smoothing with a 6-mm full width at half maximum (FWHM). These procedures were carried out using the default parameters of SPM12, with the exception of the reference time for slice-timing correction and the FWHM size for spatial smoothing. After preprocessing, we identified 10 participants who had significant (> 1 mm) and/or erratic head movements; they were excluded from the analysis.

### General linear model analysis

We conducted both individual-level and group-level general linear model (GLM) analyses. In the individual-level analysis, we created a design matrix with 18 regressors. The first regressor represented the timing of the DI offer onset. The second to fourth regressors served as parametric modulators associated with the first: DI, SR, and response time at each onset. The fifth regressor indicated the timing of the AI offer onset. The sixth to eighth regressors served as parametric modulators associated with the fifth: AI, SR, and response time at each onset. These parametric modulators were not orthogonalized with respect to each other. The nineth through eleventh regressors represented the timings of the proposer's name and face presentation, the button press, and feedback, respectively. The 12th through 17th regressors modeled the participant's head movements, while the 18th regressor was a constant term.

The event durations for the first to eighth regressors were set to the participant's response time for each trial, while the nineth to 11th regressors had an event duration of 0 seconds. We performed the GLM analysis with these design matrices using the following default settings: a high-pass filter cutoff of 128 s and a canonical hemodynamic response function for the basis function.

For the group-level analysis, we used contrast images related to the parametric modulator of DI from the individual-level analysis. We constructed a factorial design to explore neural activity associated with DI (GLM1) and analyzed the results with cluster-level FWE correction at $p < 0.05$ (with a cluster-forming height threshold of $p < 0.001$). We additionally created three multiple regression models to compare neural activity related to DI and other computational-model parameters. These regression models included the participant's $\beta(DI)$ and $\beta(SR)$ (GLM2), $\gamma(DI)$ and $\gamma(SR)$ (GLM3), and $\beta(DI)$ and $\gamma(DI)$ (GLM4), respectively (including the intercept). We investigated the results using cluster-level FWE correction at $p < 0.05$ (with a cluster-forming height threshold of $p < 0.001$). Similarly, we created a factorial design to explore neural activity related to SR. We constructed the second-level regression model, which included $\beta(SR)$ and $\beta(DI)$ based on the first-level results with SR (GLM5) and an intercept.

### Functional connectivity analysis

To uncover the neural mechanisms for accepting DI offers, we examined functional connectivity using seed regions derived from the peaks identified in Fig 4A. This analysis was conducted using the generalized psychophysiological

interaction (gPPI) toolbox [39] in SPM12. We first extracted the BOLD time series from individual-level volumes of interest (VOIs) identified in the GLM1 results. Each VOI was defined as a sphere with a 3-mm radius centered on the identified peak positions. The gPPI toolbox constructed PPI variables by multiplying the VOI time series with each GLM regressor (e.g., DI). Statistical significance was determined using a threshold of $p < 0.05$ corrected for multiple comparisons at the cluster-level, with a cluster-forming height threshold of $p < 0.001$.

To investigate regulation from the dACC, we created an individual-level gPPI design using a seed VOI at coordinates [−10, 34, 32] (the main peak of the dACC cluster in Fig 4A). The GLM design matrix included 30 regressors. The first through 11th were similar with the regressors in the GLM1 design. The 12th through twenty second regressors were the convolution of the seed VOI time series into each regressor of GLM1 (representing the multiplications of the VOI time series and each GLM1 regressor). The twenty-third regressor was the seed VOI time series. The twenty-fourth through 20 nineth regressors modeled the participant's head movements. The 13th regressor was a constant.

The event durations for the first to eighth and 12th to 19th regressors were set to the participant's response time for each trial, while the nineth to 11th and 20th to twenty second regressors had an event duration of 0 s. In this analysis, we contrasted the 13th regressor (representing DI) in the individual-level analysis using the one-sample $t$ tests with the negative contrast (i.e., −1) in the group-level analysis to represent regulation. Thus, we specifically tested how connectivity between the seed and target regions was modulated by the magnitude of DI at the offer onset.

Next, we repeated the same gPPI procedure to verify whether the vlPFC showed functional connectivity with the amygdala when DI was large. For this, we constructed an individual-level PPI design using a seed ROI at coordinates [38, 44, 0] (the main peak of the vlPFC cluster in Fig 4D) multiplied by DI. Individual-level gPPI analyses were performed to identify regions showing synchronized activity with the vlPFC at the time of the offer when DI was large. A group-level PPI analysis was conducted using one-sample $t$ tests with positive contrasts (i.e., 1).

Given our priori hypothesis of interaction between the vlPFC and amygdala, we created target masks for the left and right amygdala using the Harvard-Oxford Subcortical Structural Atlas, applying an 80% probability threshold for voxel inclusion in the amygdala. Small volume correction for multiple comparisons was applied (cluster-level FWE corrected, $p < 0.05$). In this analysis, two participants were excluded due to signal dropout in their amygdala regions on fMRI images.

We investigated the relationship between functional connectivity and behavior. However, because raw response times may not be suitable for group-level neural activity considering individual differences, we also prepared normalized response times by subtracting the mean response time of the SR/OR ratio = 1:1, where the response was not affected by DI or AI, from the raw response time for each participant.

## BOLD signal time series analysis

We obtained BOLD signal time series by calculating a PSTH at the provided position. We obtained each participant's PSTH at the dACC (MNI coordinates are [−10, 34, 32]) from the main GLM result. We calculated the mean of the PSTHs across 31 and 32 participants with high and low $\beta$(DI), respectively.

## Supporting information

**S1 Fig. Relationship between drift parameters and simulated behavior. (A)** $\beta$(DI) and rejection rate in DI conditions. The simulated rejection rate increased with increasing $\beta$(DI) (Pearson's $r = 0.776$, Holm-Bonferroni corrected $p = 3.79 \times 10^{-8}$). **(B)** $\beta$(DI) and simulated mean response time for acceptance and **(C)** rejection in DI conditions. The simulated mean response time for acceptance had no correlation with increasing $\beta$(DI), while the simulated mean response time for rejection decreased with increasing $\beta$(DI) ($r = -0.556$ and Holm-Bonferroni corrected $p = 9.80 \times 10^{-5}$, respectively). **(D)** $\beta$(SR) and simulated rejection rate. The simulated rejection rate decreased as $\beta$(SR) increased (Pearson's $r = -0.644$, Holm-Bonferroni corrected $p = 3.79 \times 10^{-8}$). **(E)** $\beta$(SR) and simulated mean response time for acceptance and **(F)** rejection. The simulated mean response time for acceptance decreased with increasing $\beta$(SR) (Pearson's $r = -0.409$,

Holm-Bonferroni corrected $p = 1.73 \times 10^{-3}$), but the simulated mean response time for rejection had no correlation. **(G)** $\beta$(AI) and simulated rejection rate. The simulated rejection rate decreased as $\beta$(AI) increased (Pearson's $r = -0.331$, Holm-Bonferroni corrected $p = 0.0242$).
(TIF)

**S2 Fig. Parameter recovery tests for varied drift parameters.** We examined the parameter ranges that can be accurately estimated by our model. **(A)** Simulated reward-seeking agents with variable $\beta$(SR). We evaluated the recovery of $\beta$(SR) while keeping the boundary, bias, nondecision time, $\beta$(DI), and $\beta$(AI) at the displayed typical values. The recovered $\beta$(SR) plateaued when its true value exceeded 3.00. Therefore, this does not pose a practical problem, as the maximum $\beta$(SR) observed across participants was 2.12. Other parameters were estimated accurately, as shown in Fig 2F. We then simulated inequity-aversive agents with variable $\beta$(SR), $\beta$(DI), and $\beta$(AI) through B to D. We assessed the recovery of $\beta$(DI) and $\beta$(AI), while varying $\beta$(SR) within a reasonable range (0.00 in **B**, 0.93 in **C** and 2.12 in **D**). The recovered $\beta$(DI) and $\beta$(AI) also plateaued when their true values were larger than 3.00. Again, this does not present a practical issue, as the maximum $\beta$(DI) and $\beta$(AI) observed across participants were 1.58 and 2.39, respectively. Other parameters were estimated accurately, as in Fig 2F.
(TIF)

**S3 Fig. Neural activity correlated with DI, which was controlled for choice and response time.** Consistent neural activity in the dorsal ACC was found (MNI coordinate = [−4, 36, 12]; cluster-level family-wised error (FWE) corrected $p < 0.05$). All other activities are listed in S3 Table.
(TIF)

**S4 Fig. Peak coordinate locations of the vlPFC cluster identified in the PPI analysis.** (A) The vlPFC cluster and peak position ([38, 44, 0] in MNI coordinate) reported in the main text, (B) the Brodmann areas surrounding the cluster peak coordinate (only the right hemisphere labels are shown), and (C) the standard brain used in MRIcroGL software are displayed. Each image is overlaid with a crosshair showing the peak. The vlPFC cluster peak lies in Brodmann area 47 and is situated within gray matter.
(TIF)

**S5 Fig. Amygdala activation in response to DI in the half of participants ($n = 31$) with high $\beta$(DI)s.** First-level results from GLM1 were used to conduct a second-level analysis averaging across these participants. Because DI-correlated amygdala activity was reported, we applied small volume correction using amygdala ROI applying an 80% probability threshold from the functional connectivity analysis (see Materials and methods section). Significant activity in the amygdala was found (cluster-level FWE corrected $p = 0.0141$, $t = 3.43$, cluster size $= 16$ mm³; peak MNI coordinates [−26, −2, −18]).
(TIF)

**S6 Fig. Neural activity correlated with SR weighted by–$\beta$(SR).** Nonsignificant neural activity in the ACC was identified (Peak MNI coordinate = [0, 26, 28]; uncorrected $p < 0.01$).
(TIF)

**S7 Fig. Comparison of $\beta$(DI) and $\gamma$(DI) for detecting dACC activity.** $\beta$(DI) and $\gamma$(DI) were contrasted to compare relative power to detect activity in the dACC. Two clusters in the dACC were found (uncorrected $p < 0.05$).
(TIF)

**S8 Fig. Correlation of boundary and bias parameters with behavior. (A)** Boundary and rejection rate in DI conditions. The rejection rate decreased with increasing boundary (Pearson's $r = −0.504$, Holm-Bonferroni corrected $p = 2.52 \times 10^{-5}$). **(B)** Boundary and mean response time for acceptance and **(C)** rejection. The mean response time for acceptance and rejection had a significant correlation with increasing boundary (Pearson's $r = 0.5207$ and Holm-Bonferroni corrected

$p = 2.43 \times 10^{-5}$, $r = 0.7190$, and Holm-Bonferroni corrected $p = 8.44 \times 10^{-9}$, respectively). **(D)** Bias and rejection rate. The rejection rate decreased as bias increased (Pearson's $r = -0.634$, Holm-Bonferroni corrected $p = 0.0107$). In addition, no correlation between bias and the mean response time for acceptance and rejection was found.
(TIF)

**S9 Fig. Coverage in EPI images.** The mean normalized EPI image across all participants and the normalized T1 structural image are displayed side by side in 3-mm slices from the orbitofrontal cortex to the amygdala. (A) Sections at $y = 44$ and $z = -24$ in MNI coordinates. (B) Section at $y = 24$ and $z = -21$ in MNI coordinates. (C) Section at $y = 4$ and $z = -18$ in MNI coordinates.
(TIF)

**S1 Data. Actual and simulated behaviors.** (A) Actual rejection rates across all trials. (B) Rejection rate in simulation recovered from estimated DDM parameters. (C) Actual mean response times for each choice and reward ratio. (D) Mean response times in simulation.
(XLSX)

**S2 Data. Model selection results calculated by WAIC.** "Model" in the data represents a combination of variables in drift term.
(XLSX)

**S3 Data. Evaluation results of generalization performance.** "Trial type" represents a reward ratio. For example, Trial type 0 corresponds to reward ratio of 1/9, and Trial type 4 corresponds to reward ratio of 1/1.
(XLSX)

**S4 Data. Parameter recovery tests.** We illustrate simulated reward-seeker (A) and inequity avoider (B). (C)–(F) demonstrate how correctly parameters could be recovered when we varied them.
(XLSX)

**S5 Data. Relationship between DDM parameters and behaviors.** (A) actual behavior and DDM parameters. (B) simulated behavior and parameters.
(XLSX)

**S6 Data. Relationship between neural activity, DDM parameters, and behavior.** (A) SPM betas correlated with DI at each peak position. (B) Signal increase from offer onset in low $\beta$(DI)s and high $\beta$(DI)s. (C) SPM betas of the PPI analysis from the dACC ROI at each peak position. (D) PPI analysis results and behavior.
(XLSX)

**S7 Data. SPM betas correlated with DI under controlling for choices and response times.**
(XLSX)

**S8 Data. SPM betas of the PPI analysis from the vlPFC ROI.** (A) Connectivity between the vlPFC and the amygdala (cluster-level FWE corrected $p < 0.05$). (B) SPM betas extracted from the amygdala activation in response to DI in the participants with high $\beta$(DI)s (cluster-level FWE corrected $p = 0.0141$).
(XLSX)

**S9 Data. Neural activity correlated with SR.** See also S6 Fig (uncorrected $p < 0.01$).
(XLSX)

**S10 Data. SPM betas from the peak correlated with the coefficient of the standard value-based model.** (A) SPM betas extracted from the dACC (uncorrected $p < 0.005$) (B) Comparison result of $\beta$(DI) and $\gamma$(DI) for detecting dACC activity (uncorrected $p < 0.05$).
(XLSX)



**S1 Table. Evaluation of each parameter's contribution to actual choices based on the generalized linear-mixed regression.**
(XLSX)

**S2 Table. Evaluation of each parameter's contribution to simulated choices based on the generalized linear-mixed regression.**
(XLSX)

**S3 Table. Neural activity correlating with DI, which was controlled for the choice and response time.** Cluster-level family-wised error (FWE) corrected $p < 0.05$.
(XLSX)

**S4 Table. Neural activity correlated with DI in the half of participants with high $\beta$(DI).** Small volume correction for multiple comparisons was applied (cluster-level FWE corrected, $p < 0.05$).
(XLSX)

**S5 Table. Neural activity correlating with SR weighted by–$\beta$(SR).** Statistical threshold for significance at $p < 0.01$ uncorrected for multiple comparisons. Clusters larger than $8\,mm^3$ are displayed.
(XLSX)

**S6 Table. Neural activity identified by the PPI analysis whose seed ROI is the ACC in S1 Table.** Statistical threshold for significance at $p < 0.001$ uncorrected for multiple comparisons. Clusters larger than $8\,mm^3$ are displayed.
(XLSX)

**S7 Table. Neural activity correlating with SR weighted by $\beta$(SR).** Statistical threshold for significance at $p < 0.001$ uncorrected for multiple comparisons. Clusters larger than $8\,mm^3$ are displayed.
(XLSX)

**S8 Table. Results of robust linear regression on the mean rejection rate.**
(XLSX)

**S9 Table. Results of robust linear regression on the mean normalized response time for acceptance.**
(XLSX)

**S10 Table. Task parameter set.**
(XLSX)

**S11 Table. Variance inflation factor (VIF) for each task parameter.**
(XLSX)

## Acknowledgments

We thank Satoshi Tada, Koji Fuji, and Michiko Hattori for their generous technical support, as well as Peter Karagiannis for editing drafts of the manuscript.

## Author contributions

**Conceptualization:** Shotaro Numano, Masahiko Haruno.

**Formal analysis:** Shotaro Numano, Masahiko Haruno.

**Funding acquisition:** Masahiko Haruno.

**Investigation:** Shotaro Numano, Chris Frith, Masahiko Haruno.

**Project administration:** Masahiko Haruno.

**Supervision:** Masahiko Haruno.

**Validation:** Shotaro Numano, Masahiko Haruno.

**Visualization:** Shotaro Numano.

**Writing – original draft:** Shotaro Numano.

**Writing – review & editing:** Chris Frith, Masahiko Haruno.

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
