## [Editor Report · Decision Letter 0]

8 Jan 2025

Dear Masahiko,

Happy New Year!

Thank you for submitting your manuscript entitled "Overcoming pride via the dorsal ACC underlies acceptance of unfair offers" for consideration as a Research Article by PLOS Biology.

Your manuscript has now been evaluated by the PLOS Biology editorial staff as well as by an academic editor with relevant expertise and I am writing to let you know that we would like to send your submission out for external peer review.

Once your full submission is complete, your paper will undergo a series of checks in preparation for peer review. After your manuscript has passed the checks it will be sent out for review. To provide the metadata for your submission, please Login to Editorial Manager (https://www.editorialmanager.com/pbiology) within two working days, i.e. by Jan 10 2025 11:59PM.

Kind regards,

Christian

Christian Schnell, PhD

Senior Editor

PLOS Biology

cschnell@plos.org

---

## [Decision Letter · Decision Letter 1]

10 Mar 2025

Dear Masahiko,

Thank you for your patience while your manuscript "Overcoming pride via the dorsal ACC underlies acceptance of unfair offers" was peer-reviewed at PLOS Biology. It has now been evaluated by the PLOS Biology editors, an Academic Editor with relevant expertise, and by several independent reviewers.

In light of the reviews, which you will find at the end of this email, we would like to invite you to revise the work to thoroughly address the reviewers' reports.

As you will see below, the reviewers think that your study is interesting, but they also raise a lot of concerns, for example about the statistical analyses, overinterpretation of the results, lack of clarity, unclear conclusions and missing analyses. Please carefully take the reviewers comments and suggestions regarding additional analyses into account when revising your manuscript.

Given the extent of revision needed, we cannot make a decision about publication until we have seen the revised manuscript and your response to the reviewers' comments. Your revised manuscript is very likely to be sent for further evaluation by all or a subset of the reviewers.

**IMPORTANT - SUBMITTING YOUR REVISION**

*Re-submission Checklist*

*Published Peer Review*

*PLOS Data Policy*

*Blot and Gel Data Policy*

Sincerely,

Christian

Christian Schnell, PhD

Senior Editor

PLOS Biology

cschnell@plos.org

REVIEWS:

Reviewer #1: This is an innovative study of an understudied but important area of cognitive neuroscience, namely how to understand the motivations underpinning (altruistic) punishment and their neuroanatomy.

My main concerns are the following:

1) The interpretation of results as reflecting suppression of emotions is not being directly tested by the experiment and therefore a descriptive rather than mechanistic conceptualisation should be used with mechanistic interpretations discussed as potential explanations for the findings:

1a) Abstract, Intro and Discussion: "suppressing" emotions is not really measured as it could also be that the anger driving inequity aversion is replaced by another emotion, e.g. the anticipated monetary reward value of accepting the offer. I would therefore use a more descriptive term instead of "suppression", such as "overcoming" or "replacing" or "deciding against"

1b) at the end of the introduction, long response times are equated with suppressing negative feelings, but an equally likely and more plausible explanation in my view would be higher levels of choice ambivalence which would also align well with dorsal anterior cingulate activations when conflicting choices are introduced (Botvinick et al), as well its more general role in salient feelings with behavioural relevance such as social and physical pain (Eisenberger's work).

2)The computational analysis methods are sophisticated and I am not an expert on computational modelling. Nevertheless I would expect the authors to explain their methods in a way which can also be understood by someone who is only familiar with standard statistical approaches. From a standard statistical perspective, there were several concerns:

2a) It appears that the authors first optimized their models as presented in the model selection results and then used the best model for their further analyses, I am only familiar with prediction modelling where this would be seen as inflating the significance of the final analysis (double-dipping), could the authors address this please?

2b) There were multiple statistical results presented, but no multiple comparison correction.

3) Terminology - and connection with the wider literature: "inequity aversion" should be linked with the existing literature on altruistic punishment and indignation / moral anger

4) As standard fMRI sequences lead to large signal loss in brain regions relevant for emotions such as the orbitofrontal cortex, it will be important to discuss this in the conclusions and add images of the implicit mask produced by SPM to the supplement to show the coverage of orbitofrontal areas and describe the areas not covered.

Minor comments:

5) abstract: was it bilateral vlPFC?

Reviewer #2: Summary: This study aims to understand factors promoting decisions regarding unfair offers in the Ultimatum Game. It addresses an interesting question: what leads people to accept these offers? While most research has tended to address why such offers are rejected, the authors rightly point out that less has been done to understand when and why people are willing to accept, arguing that the primary explanations put forth so far tend to center around self-interest, ignoring other potential factors. In their study, the authors claim to address the extent to which overcoming pride through emotion regulation might play a role. To study this, they examine behavior of participants accepting or rejecting Ultimatum Game offers. They decompose this behavior using the drift diffusion model (DDM) using several parameters, including weights on self-interest, disadvantageous inequality (DI), and advantageous inequality, a starting bias, threshold, and non-decision time. They find that the model reproduces key behavioral findings, including increasing rejection with greater DI, as well as a tendency to take longer to accept rather than reject extremely unfair offers. They tie this to weight on DI, showing that individuals with lower weight on DI show GREATER correlations between DI and anterior cingulate responses. In turn, they show that the ACC is negatively functionally connected with the vlPFC when DI is large, and that the more this is true, the less DI is rejected. They also show that the vlPFC shows some functional connectivity with the amygdala, though this connectivity shows less clear relationships to choice behavior.

Overall evaluation: While there are elements to like about this study, including a relatively large sample size, a computational approach, and an interesting question, I found my enthusiasm tempered by several factors. Perhaps largest among these is the overinterpretation of results, largely based on reverse inference, showing that the ACC is necessary to overcome pride, or otherwise is performing an emotion regulation role. I also felt that at a number of points there is a lack of clarity about key methods that made it hard to know exactly what the authors did, and what they found. I also think the authors need to do much more to exclude alternative explanations for their results. Finally, while I think the computational modeling approach is admirable, I think much more could and should be done to unpack these results and to understand what they mean. What follow are my more specific points on each of these issues.

1. My biggest concern in the paper is what feels like an attempt to "oversell" the results. The title of the paper claims to show that the ACC is necessary to overcome pride in the Ultimatum Game. The introduction and discussion similarly suggest that what the authors have shown is that the interactions between ACC, vlPFC, and amygdala are necessary for "suppressing the negative response to inequity in the amygdala" and that these results may contribute to understanding well-being. I don't think such claims are well-justified given the design of the study. At no point is pride, or any other emotion, measured. Nor is there really any way to know what role the ACC is performing in this study (see my comments below). Instead, the paper engages in quite a lot of reverse inference to arrive at a connection between pride, negative emotion, and emotion regulation. Given that the ACC region the researchers report is frequently observed for all sorts of conflict-related tasks and its function is hotly debated, I didn't feel confident that the authors' interpretation was justifiable. They'd likely need to tone down their interpretation considerably, and stick much closer to the data.

2. My second, related issue concerns a set of alternative interpretations of the results that are perhaps much less 'interesting' but more plausible, in my view. First, research suggests that areas related to conflict and cognitive control, including the ACC, may respond more strongly when there is greater conflict, and when RTs are longer, and may even track the degree of evidence accumulation being performed during social decisions (Hutcherson & Tusche, 2022). The authors do not control for RTs in any of their analyses, but it is clear from their modeling that DI is likely strongly correlated with response time in this study. This is less likely to be true (or at least, could be less true) in people who are less bothered by DI (i.e., have a lower weight on this factor). Thus, one could observe a modulation of the association between DI and ACC response as a function of the weight on DI not because the ACC somehow causally reduces pride, causally regulates negative emotion, or even causally changes rejection rates, but simply because other factors shaping these things could increase (or decrease) the relative conflict (or RT) associated with high and low-DI trials. This could similarly explain the connectivity results. Without reporting a much more extensive assessment of how DI (or SR) correlates with trial-by-trial response times and response conflict (either in observed behavior, or in simulations from DDMs), and showing that this does not account for the pattern of results, I have difficulty trusting the authors' preferred interpretation. At minimum, I would want to know whether these results survive after controlling for RTs. I'd also want to know whether the ACC region the authors are finding shows evidence that it is simply tracking RTs or simply correlating with overall evidence accumulation signals.

3. The authors could also provide much more detail on the DDM results themselves. First, 500 burn in and 500 samples seems like a relatively small amount of fitting for an hSSM model. It is comforting to see the model produce aggregate level results that capture basic patterns, but that is a far cry from saying that the model fits well and converges to a point where we can have confidence in the estimated values of individual-level parameters. Can the authors report parameter recovery statistics, and convergence statistics of the winning model (e.g., Gelman-Rubin diagnostics for all parameters). Second, the authors never provide any details about what the values of the best-fitting parameters are. This would provide a lot of interesting information for the reader, such as how self-interest vs. DI vs. AI are weighted, whether there is a starting bias to reject (or accept) offers overall, and whether things like the threshold might also be correlated with RTs (it should be). I'd be particularly interested to know whether and how thresholds and starting biases correlate with any of the observed effects.

4. The authors are similarly elliptical about the fMRI analyses. Although I think I understood the basics of their primary GLM, I was left with a number of questions about the PPIs that they report. They mention performing a gPPI, which I take to mean that they included all original regressors from the primary GLM, and then examined the neural activity associated with the PPI as well, but it is not 100% from their methods whether this is the case. Similarly, it was unclear to me whether ALL PPIs (including the ACC and vlPFC PPIs) were essentially interacting the neural signatures with the parametric regressor for DI. For instance, for the vlPFC connectivity, sometimes they simply say that they looked at connectivity "at offer time", and at other times it sounds as if they are looking at connectivity modulated by DI. In general, being clearer about what was done would be helpful.

5. The authors argue that accepting unfair offers takes more "regulation" than rejecting them, but all of the analyses they report where neural activity or connectivity is correlated with RTs involves simply the raw response time, rather than the response time difference to accept vs. reject. There's a lot that can go into a raw response time, including a participant's simple motor speed, the threshold they've got for a response, they're starting bias, etc. None of these have to do with overcoming negative emotion or pride. So I'm not clear on why the raw response time is the appropriate measure here, rather than the difference between acceptance and rejection. Do the results hold if that difference is used instead of the raw RT?

6. The authors never report basic neural correlates of parametric modulators for SR, DI, or AI, so it is not clear whether the observation of ACC modulation and connectivity, or of vlPFC to amygdala connectivity modulated by DI has any real relevance to DI processing. This is especially problematic for the author's conceptual diagram, which suggests that ACC modulates vlPFC which modulates amygdala and then alters choice. If the amgydala isn't correlating with DI to begin with, or if its response to DI is not correlated with the weight on DI, then simply observing connectivity to vlPFC (which itself isn't obviously correlated with DI, or with accepting or rejecting offers) doesn't tell you that the vlPFC is somehow regulating negative emotion, directed by the ACC. Amygdala connectivity with the vlPFC is also only identified using small-volume correction with a fairly liberal threshold, and does not correlate with rejection rates (though does correlate with raw response times) so the result feels a little bit like it was manufactured so as to draw a connection to emotional processing, rather than like it reflects robust evidence for the authors' model.

7. The vlPFC region highlighted in Figure 3 looks like it might mostly be in white matter - can the authors comment on this? This isn't the region I would normally consider "vlPFC" in a regulatory sense, so I'm not sure how to think about what this region is.

8. The DI regressor in the GLM should be 0 any time there is a fair or advantageous offer. This raises the concern for me that any evidence for differences as a function of DI (especially for PPIs) is largely just tracking whether the offer is 'easy' to accept in the sense of being fair, and thus likely to have a fast RT. If the authors ran a GLM in which there were two onset regressors, one for fair/advantageous trials and one for unfair/disadvantageous trials, with the latter parametrically modulated by the degree of DI, do the results look similar? This would be a more convincing test fo the idea that it is sensitivity to the DEGREE of DI that is driving results, rather than just the presence of inequity per se.

Reviewer #3: This study investigates the neural correlates of accepting unfair offers during the ultimatum game, thereby complementing many studies that focused more on the neural mechanisms involved in rejection of unfair offers. The authors describe a (standard) fMRI study of the ultimatum game and model rejection behavior with a drift-diffusion model (DDM) that incorporates the level of self-reward and inequality as modulators of the drift rate. The model captures observed behavior, and the level of disadvantageous inequality correlates with ACC activity across trials; moreover, this effect appears stronger in people who accept unfair offers more often. Connectivity analyses furthermore show that functional interactions between ACC and vlPFC were more negative when DI was large, and that vlPFC-amygdala connectivity correlated negatively with rejection rates and response times for acceptance decisions, suggesting that a functional ACC-vlPFC-amygdala circuit is involved in the suppression of internal (affective?) states related to accepting unfair offers.

The study is interesting in that it examines the acceptance of unfair offers, a behavior that has been much less studied than the rejection of such offers. In general, the study design and methods are solid, and the results should be of interest to the social neuroscience community. However, I have several concerns about the analyses and results that the authors would need to address before their results can be considered conclusive.

(1) Initial behavioral analyses: The authors describe that they varied the payoffs from trial to trial by monetary amounts that are not insubstantial (especially compared to the smaller payoffs). However, the initial analyses confirming the behavioral effects of inequality are rather crude: They comprise overall ANOVAs followed by t-tests between the most extreme levels of inequality. This is not convincing; moreover, comparisons of cherry-picked trials (only the most extreme levels) are very prone to selection biases. Please include detailed analyses of how inequality levels affect behavior on all trials, for example regression analyses incorporating the actual payoffs/inequality values. Please do this for both the observed data and the posterior predictive data, to back up your conclusions about the behavioral effects. Also, please report descriptive and statistical analyses of the DDM parameters (e.g., which ones are different from zero?), since these can tell the reader a lot about the behavioral effects of inequality levels on both choices and response times when considered together.

(2) Specificity of the effects: The authors interpret their results as if the neural activity reflected an emotional mechanism that is uniquely expressed for acceptance of unfair offers. However, the neural analyses link brain activity to the strength of disadvantageous inequality across all trials, irrespective of whether participants accepted or rejected. Please compute GLMs that include choice as a variable and show that the ACC activation is specific for the acceptance of unfair offers, not just correlated with the level of disadvantageous inequality irrespective of choice. Moreover, I wondered to what degree the results generally reflect the level of conflict a participant experiences, rather than specific emotions associated with accepting unfairness? The authors should report whether this is the case, for example by including response time as a covariate in the GLM and examining whether the ACC activation is still present when controlling for general conflict. In any case, this issue should be discussed.

(3) Selective analysis of only parts of the behavioral data: In the analyses presented in Figure 2, the authors report results for "the DI condition". Please report the corresponding results also for the other conditions, and whether the results are specific for DI trials.

(4) Selective analysis of only parts of the neural data: Likewise, the analyses presented in Figure 3B are very cherry-picked: They reflect data from a subset of participant selected to show specific effects, and are re-analyses of hand-selected parts of the hemodynamic response for effects used to initially define the ROI. This is bad research practice (double-dipping and cherry-picking). Please report the corresponding analyses as covariations across all participants (with scatterplots), and use temporally unbiased methods to analyze the hemodynamic response.

(5) Separation of SR, DI, and AI: The GLM contains SR, DI, and AI as parametric modulators. Are they correlated? Can they really be dissociated? The model suggests that, but please report their actual correlations, and the methods you used to make sure the GLM captures only the unique variance (regressor othogonalization).

(6) Behavioral comparison between logistic regression and DDM: It is not surprising that only the DDM but not the regression can capture reaction times, since response times are only used to fit the DDM and do not enter the regression model. This is a feature of the models, not a "better or worse performance". Please discuss this appropriately in the manuscript. Also, please use formal model comparison to see whether for the choice data (acceptance rates) alone, the DDM does better than the logistic regression. Otherwise it is much less justified to say that the logistic performs poorly at capturing the behavior - it may do well at what it is set up to do (explaining choices), but may just not capture response times since these are not included in the model. In any case, the results description and discussion should reflect this qualitative difference between the models.

(7) Neural comparison between logistic regression and DDM: The authors claim that "a conventional value-based system (i.e., logistic regression) that only considers choices performed poorly compared with the DDM in detecting these fMRI signals". They base this on higher p-values for the same contrast, but never directly compare whether the DDM does better. Please include such a comparison, for example by Bayesian fMRI model comparison or via a GLM that adds both sets of regressors (DDM and logistic) and examines whether the DDM captures unique variance.

(8) There are many grammatical errors and typos in the manuscript, please proof read it again.

---

## [Decision Letter · Decision Letter 2]

6 Oct 2025

Dear Masahiko,

Thank you for your patience while we considered your revised manuscript "Dorsal anterior cingulate regulates inequity aversion and facilitates acceptance of unfair offers" for consideration as a Research Article at PLOS Biology. Your revised study has now been evaluated by the PLOS Biology editors, the Academic Editor and two of the original reviewers.

In light of the reviews, which you will find at the end of this email, we are pleased to offer you the opportunity to address the remaining points from the reviewers in a revision that we anticipate should not take you very long. As you will see, Reviewer 2 continues to be supportive of the manuscript but has raised some concerns about two issues and we ask that you consider these carefully in preparing a revised manuscript. One of Reviewer 2’s points concerns refraining from framing the reporting of correlational results in a causal way. The second point concerns the model recovery and asks that model recovery information is made available in a more extensive manner in the supplementary information. While realizing that is not always shown in every manuscript, the reviewer has argued that it might be especially useful in the current case.

We will then assess your revised manuscript and your response to the reviewers' comments with our Academic Editor aiming to avoid further rounds of peer-review, although we might need to consult with the reviewers, depending on the nature of the revisions.

**IMPORTANT - SUBMITTING YOUR REVISION**

*Resubmission Checklist*

*Published Peer Review*

*PLOS Data Policy*

*Blot and Gel Data Policy*

Sincerely,

Christian

Christian Schnell, PhD

Senior Editor

PLOS Biology

cschnell@plos.org

REVIEWS:

Reviewer #1 (Roland Zahn signed his report): The authors have addressed all my concerns. I have no further comments.

Reviewer #2: I appreciate the work the authors have done to address my previous comments, and I think the paper is much clearer and improved as a result. However, I still had some lingering concerns/suggestions:

1. I appreciated the authors work to address the request for parameter recovery, but I'm afraid that I don't think they've quite satisfied it. Typically, a parameter recovery analysis would examine many iterations of variation across multiple parameters, not just two, and report not only on whether the model seems to get in the ballpark for a given parameter, but also actual correlations and plots of actual vs. estimated parameters, where the ideal is dots falling on a 45 degree line. The reason this is important is because such examination can indicate places and/or parameter values for which the model breaks down, and it can also, crucially, indicate places where parameters 'trade off' against each other (e.g., where a difference in the weight given to, say, DI, is actually picked up in the model by a differences in the starting bias). I don't think the authors need to put this analysis in the main body of the paper, as I think it would be distracting, and their current approach suggests that they would likely find adequate recovery. Perhaps this view is not shared by others, but I would be much more comfortable with the model ability to fit behaviour than is currently the case seeing just two different simulated individuals fit multiple times.

2. I am still uncomfortable with the causal language the authors use, e.g., "Participants who, IN ORDER TO ACCEPT unfair offers suppressed disadvantageous inequity (DI-driven rejection". I think what the authors show here is interesting, but I am still not convinced they can really make the claim that somehow the ACC is SUPPRESSING DI-driven rejection. It is one possible interpretation, and it might be the right one, but there's no causality here. The authors show a correlation, such that ACC representations of DI are stronger when DI is represented LESS in behaviour. They also show some suggestive connectivity analyses (more on this below). But they don't, for instance, MANIPULATE ACC activation through stimulation and show that rejections of inequality go down (or up). Especially since the authors proposed causal ordering (ACC  vlPFC  amygdala) implies that amygdala sensitivity to DI should MEDIATE ACC relationships to DI, and the authors don't show this. I think my proposed solution is for the authors to simply soften their language to admit uncertainty, e.g., "Participants who had lower levels of behavioural disadvantageous inequality aversion actually showed stronger representations of that inequality in the ACC, suggesting that it may play a role in regulating subjective weights." It's a subtle difference, and I'm sorry to keep coming back to this. But the paper is making claims that are just not supportable. What you can show is interesting and suggestive. I'd prefer to just see it described as such, and acknowledged that future work will have to establish causality (which the authors already do).

3. I'd still love to see the authors visualize the negative correlation between ACC representations of DI and behavioural sensitivity to DI differently: can they simply show a scatterplot/correlation plot, where the X-axis is behavioral sensitivity B(DI) and the Y-axis is the estimated beta-coefficient for the ACC response to DI? This would really help to fully interpret ACC responses.

4. I'm not wholly sure what to make of the amygdala findings as currently reported. It is interesting that amygdala correlates with parametrically with DI in the half of participants who have the strongest B(DI), but that isn't the same thing as saying that, in the full sample, amygdala representations of DI correlate with B(DI) significantly, or that a between-groups t-test of amygdala DI representations are actually different between the high-DI and low-DI group. Maybe this is an issue of noise, since amygdala signals can be harder to measure cleanly perhaps, but I still have a hard time interpreting the relative lack of correlation with individual differences in either vlPFC-amygdala connectivity, or amygdala responsivity, in the story where the amygdala is supposedly the downstream end of a path from ACCvlPFCamgydala. Can the authors comment on what they think is going on here? Could it actually be the reverse directionality?

5. Can the authors clarify whether parametric modulators were or were not orthogonalized with respect to each other, and if so, whether results change if SR is the first variable included in the GLM. This isn't critical, I don't think, but will help to interpret neural parametric correlations (or lack thereof).

Smaller comments

6. "The amygdala was reported to respond to inequity in people who dislike it, Line 9 on pg. 19 - can you provide a citation for this?

7. For Table 2, can you remind the reader that the upper boundary of the DDM represents rejection. It's in your methods section, but it will help to remind people here to interpret the values of the weights on different quantities.

---

## [Editor Report · Decision Letter 3]

19 Nov 2025

Dear Masahiko,

Thank you for your patience while we considered your revised manuscript "Dorsal anterior cingulate regulates inequity aversion and facilitates acceptance of unfair offers" for publication as a Research Article at PLOS Biology. This revised version of your manuscript has been evaluated by the PLOS Biology editors and the Academic Editor.

Based on our Academic Editor's assessment of your revision, we are likely to accept this manuscript for publication, provided you satisfactorily address the following data and other policy-related requests:

* We would like to suggest a different title to improve its accessibility for our broad audience: "The human dorsal anterior cingulate facilitates acceptance of unfair offers and regulates inequity aversion"

* Please include the approval/license number of the ethical approval for the experiments.

* Please include information in the Methods section whether the study has been conducted according to the principles expressed in the Declaration of Helsinki.

* DATA POLICY:

Regardless of the method selected, please ensure that you provide the individual numerical values that underlie the summary data displayed in the following figure panels as they are essential for readers to assess your analysis and to reproduce it: 1BC and 2DEFG.

* CODE POLICY

We expect to receive your revised manuscript within two weeks.

*Published Peer Review History*

*Press*

Sincerely,

Christian

Christian Schnell, PhD

Senior Editor

cschnell@plos.org

PLOS Biology

---

## [Editor Report · Decision Letter 4]

25 Nov 2025

Dear Masahiko,

Thank you for the submission of your revised Research Article "The human dorsal anterior cingulate facilitates acceptance of unfair offers and regulates inequity aversion" for publication in PLOS Biology. On behalf of my colleagues and the Academic Editor, Matthew Rushworth, I am pleased to say that we can in principle accept your manuscript for publication, provided you address any remaining formatting and reporting issues. These will be detailed in an email you should receive within 2-3 business days from our colleagues in the journal operations team; no action is required from you until then. Please note that we will not be able to formally accept your manuscript and schedule it for publication until you have completed any requested changes.

PRESS

We frequently collaborate with press offices. If your institution or institutions have a press office, please notify them about your upcoming paper at this point, to enable them to help maximize its impact. If the press office is planning to promote your findings, we would be grateful if they could coordinate with biologypress@plos.org. If you have previously opted in to the early version process, we ask that you notify us immediately of any press plans so that we may opt out on your behalf.

Sincerely, 

Christian

Christian Schnell, PhD

Senior Editor

PLOS Biology

cschnell@plos.org